# Experimental Data on the Role of Melatonin in the Pathogenesis of Nonalcoholic Fatty Liver Disease

**DOI:** 10.3390/biomedicines11061722

**Published:** 2023-06-15

**Authors:** Dimitar Terziev, Dora Terzieva

**Affiliations:** 1Second Department of Internal Medicine, Gastroenterology Section, Faculty of Medicine, Medical University, 4002 Plovdiv, Bulgaria; 2MDL “Bioiv”, Medical University, 4002 Plovdiv, Bulgaria; terzieva2006@yahoo.com

**Keywords:** nonalcoholic fatty liver disease, metabolic syndrome, melatonin, experimental data, signaling pathways

## Abstract

Despite the increasing prevalence of nonalcoholic fatty liver disease (NAFLD) worldwide, its complex pathogenesis remains incompletely understood. The currently stated hypotheses cannot fully clarify the interrelationships between individual pathogenetic mechanisms of the disease. No appropriate health strategies have been developed for treating NAFLD. NAFLD is characterized by an accumulation of triglycerides in hepatic cells (steatosis), with the advanced form known as nonalcoholic steatohepatitis. In the latter, superimposed inflammation can lead to fibrosis. There are scientific data on NAFLD’s association with components of metabolic syndrome. Hormonal factors are thought to play a role in the development of metabolic syndrome. Endogenous melatonin, an indoleamine hormone synthesized by the pineal gland mainly at night, is a powerful chronobiotic that probably regulates metabolic processes and has antioxidant, anti-inflammatory, and genomic effects. Extrapineal melatonin has been found in various tissues and organs, including the liver, pancreas, and gastrointestinal tract, where it likely maintains cellular homeostasis. Melatonin exerts its effects on NAFLD at the cellular, subcellular, and molecular levels, affecting numerous signaling pathways. In this review article, we discuss the experimental scientific data accumulated on the involvement of melatonin in the intimate processes of the pathogenesis of NAFLD.

## 1. Introduction

Nonalcoholic fatty liver disease (NAFLD) is the leading cause of chronic liver disease worldwide, affecting over 25% of the United States and global populations [1]. The disease is characterized by the accumulation of droplets of lipids and mainly triglycerides within hepatocytes (steatosis), with the advanced form known as nonalcoholic steatohepatitis (NASH) [2,3]. The latter is a clinical syndrome of steatosis and hepatic inflammation that is diagnosed via liver biopsy and subsequent histological examination, after other causes of liver disease have been excluded [3]. For the diagnosis of NASH, the establishment of three pathognomonic features on liver biopsy is important—hepatocellular ballooning, lobular inflammation, and steatosis [2]. The differentiation of NASH from NAFLD is of great importance for determining the prognosis of the disease [4]. In NASH, superimposed inflammation can lead to progressive fibrosis and increase the risks of cirrhosis, liver failure, and hepatocellular carcinoma [1,2,3]. From this point of view, it can be noted that NAFLD covers a wide range of liver pathology—from steatosis alone through steatohepatitis to liver cirrhosis and cancer [5]. NAFLD is poised to become the leading indication for liver transplantation in North America and areas of Europe [6]. According to Charlton et al. [7], NASH is the third most common indication for liver transplantation in the United States, and the frequency of NASH as an indication is steadily increasing. NASH is the only indication for liver transplantation that was seen to increase from 1.2% in 2001 to 9.7% in 2009.

Over the past few years, several hypotheses have attempted to explain the role and meaning of a number of risk factors for the development of NAFLD. Although many specific features of the disease have been elucidated, there are still many unanswered questions about the pathogenetic mechanisms underlying the disease. For this reason, it is still believed today that the pathogenesis of NAFLD is not completely characterized [8]. Associations between obesity, type 2 diabetes mellitus (T2DM), and fatty liver (steatosis) have long been recognized, as has the high prevalence of cirrhosis in diabetes [5]. The best-known risk factors for NASH are obesity, T2DM, and lipid abnormalities (hypertriglyceridemia) [3,5]. It is widely considered that NASH is thought to be the hepatic manifestation of metabolic syndrome [2]. A large proportion of patients with NAFLD have metabolic comorbidities, such as obesity, T2DM, hyperlipidemia, hypertension, and metabolic syndrome [9]. However, it should be noted that not all patients with these comorbidities have NAFLD/NASH, and not all patients with NAFLD/NASH suffer from one of these conditions [10]. An international panel of experts recommended that when there is evidence of hepatic steatosis coexisting with one of the criteria of overweight/obesity, presence of T2DM, and evidence of metabolic dysregulation, to use the term metabolic dysfunction-associated fatty liver disease (MAFLD) [11]. In fact, complex metabolic disorders, such as central obesity, hyperglycemia, hyperlipidemia, hypertension, insulin resistance, and hepatic steatosis, which are risk factors for cardiovascular diseases and T2DM, fill the content of the concept of metabolic syndrome [12]. In addition to these disorders and genetic factors, physical inactivity, pro-inflammatory state, and hormonal changes are believed to play a role in the development of metabolic syndrome [13]. Essential characteristics of hepatic metabolism are its high dynamics, its influence on fasting/fed state, and circadian rhythms [14]. It is believed that, on the one hand, the liver circadian clock is strongly influenced by feeding/fasting rhythms, which is associated with altered activity of genes regulating the metabolism of glucose, lipids and bile acids, autophagy, and stress of the endoplasmic reticulum. Diet and feeding/fasting rhythms also modulate the cyclical change in the gut microbiome. On the other hand, peripheral circadian rhythms, including those of the liver, are regulated by the biological clock located in the hypothalamic suprachiasmatic nucleus [15]. It can be concluded that the desynchronization of these processes, both at the central and peripheral level, would be important in a number of metabolic disorders that occupy a leading place in the pathogenesis of NAFLD.

One of the hormones that has a pronounced circadian rhythm of secretion with physiologically high values in the blood at night is melatonin. More than 20 years ago, the presence of high-affinity melatonin receptors in mouse hepatocytes, whose binding affinity to melatonin was affected by blood glucose levels, was established [16]. There is evidence that melatonin administration reduces body weight in obese laboratory rats, but melatonin efficiency was time dependent [17]. In aged obese rats, insulin sensitivity is increased after melatonin supplementation, suggesting that an age-related decline in melatonin secretion is likely to play a role in the development of insulin resistance in aged organisms [18]. The close relationship observed between insulin resistance and NAFLD is confirmed by elevated free fatty acid levels during fasting and a reduced suppression of lipolysis after insulin administration, which strongly correlate with the degree of fatty infiltration of the liver [19].

In this review article, we discuss the scientific data from experimental studies on the involvement of neurohormone melatonin in the regulation of signaling pathways that matter in the pathogenesis of NAFLD.

## 2. Melatonin, General Data, Mechanisms of Action, and Physiological Significance

Melatonin (N-acetyl-5-methoxytryptamine) is an indoleamine, mainly produced and secreted by the pineal gland during the dark phase of the day [20]. The circadian rhythm of melatonin is characterized by large inter-individual variation, but it is very robust and reproducible in the same subject. It is believed that the melatonin rhythm is not only a reflection of the day–night cycle but is internal to the organism, resulting from cyclical signals, possibly coming from the suprachiasmal nucleus [21]. In addition to light, age also affects the melatonin rhythm [22,23]. Melatonin exerts receptor-mediated and non-receptor-mediated action, and melatonin’s target sites are both central and peripheral [24]. Melatonin is a lipophilic hormone, which acts via high-affinity specific G-protein-coupled membrane receptors (MT_1_ and MT_2_) expressed in brain structures (the suprachiasmatic nucleus, the cerebellum, the cortex, the hippocampus, the hypothalamus), retina, adipose tissue, and kidney [17]. A binding site MT_3_ for melatonin was found in hamster kidney, which, after sequencing using mass spectrometry, was identified as the hamster homologue of the human quinone reductase 2 [25]. The high fat solubility of melatonin allows it to freely cross the cell and nuclear membranes [24]. It is assumed that melatonin is a ligand for the retinoid-related orphan receptor (ROR) and retinoid Z receptor (RZR) (ROR/RZR) family of orphan nuclear receptors [26]. Additionally, melatonin binds to cytosolic calmodulin and, thus, directly affects calcium signaling, protecting macromolecules (DNA) and other cellular structures from oxidative damage [27,28]. Recently, attention has been paid to the fact that melatonin presents several mechanisms of action, which result in different time-allocated effects [28]. Two types of effects of melatonin are discussed—one occurs at night, when the secretion of melatonin from the pineal gland is high (immediate effects), and the other occurs during the next light part of the day, when the secretion of the hormone is very low (prospective effects).

The primary physiological function of melatonin is to convey information concerning the daily cycle of light and darkness to body structures [20]. The circadian organization of physiological functions, for instance, immune, antioxidant defenses, haemostasis, and glucose regulation, depend also on the melatonin signal. Melatonin probably consolidates the whole rhythmic organization through its chronobiotics properties [28,29]. It is believed that through its prospective effects, melatonin can control the circadian time of the suprachiasmal nucleus, and pancreatic ß cells, islets, and the metabolically most important tissues (liver, muscle, and adipose tissue) are targets for melatonin [28]. As a strong chronobiotic, melatonin influences the circadian distribution of metabolic processes, synchronizing them to the activity–feeding/rest–fasting cycle [30]. Melatonin is an important player in the regulation of energy balance and carbohydrate and adipocyte metabolism [29,30]. Table 1 summarizes the metabolic and chronobiological effects of melatonin that influence energy metabolism and, ultimately, body weight.

Since the regulating system of melatonin secretion is complex, there are many pathological situations where melatonin secretion can be disturbed [20]. For example, with aging, night-shift work, or illuminated environments during the night, the secretion of melatonin is suppressed [30]. As a result, consequences associated with disturbances in metabolism (insulin resistance, glucose intolerance, sleep disturbance, dyslipidemia) and in metabolic circadian synchronization (chronodisruption) occur, which can lead to metabolic disorders and obesity. It is assumed that the use of melatonin replacement therapy may protect and/or help eliminate these pathologies. The effects of melatonin administration (in experimental or clinical studies, in treatment) depend on a number of factors, such as time and route of administration, concentration and duration of administration, regularity of intakes, and specificity of the target organ (for example, presence or absence of different melatonin receptors) [28]. Ectopic fat accumulation, particularly in the liver, is frequently observed in obese persons and is strongly associated with metabolic dysfunction, including multiorgan insulin resistance and dyslipidemia [31]. The possibility of interaction between exercise and diet on the mechanisms that regulate liver fat accumulation and depletion is discussed. Obesity is a multifactorial disease (it is not just increased food intake that matters), leading to difficulties in correcting metabolic disorders [29]. One population-based propensity score-matched study evaluated the association between circadian misalignment and MAFLD and found that the prevalence of MAFLD was higher in the circadian misalignment group than the non-circadian misalignment group (45.41% vs. 28.41%, *p* < 0.001) [32]. The data also suggest that the presence of circadian misalignment increased the risk of MAFLD by more than twofold and that circadian misalignment is independently associated with MAFLD, while short sleep duration alone (<6 h) is not independently associated with this risk. The authors concluded that in addition to diet, exercise, and pharmacological therapy, it is appropriate to make efforts to improve existing sleep or chronotype disorders. In a study by Wyatt et al. [33], oral melatonin (0.3 mg or 5.0 mg) or identical-appearing placebo was administered 30 min prior to each sleep episode during forced desynchrony in 36 healthy men and women, between the ages of 18 and 30. According to survey data, the administration of exogenous melatonin in young men and women has circadian phase-dependent hypnotic properties that enhance sleep consolidation occurring outside the period of endogenous melatonin secretion. Results support the hypothesis that both exogenous and endogenous melatonin attenuate the wake-promoting drive from the circadian system. The action of melatonin as a chronobiotic was confirmed by a study in seven totally blind people with free-runing circadian rhythms, who received 10 mg of oral melatonin or placebo nightly, one hour before their preferrable bedtime [34]. The results indicated that the phase-advancing effects of melatonin are of sufficient magnitude to entrain free-running circadian rhythms in most blind persons who have such rhythms, thereby preventing severe sleep disturbance. In a mini review by D’Souza et al. [35], the pathogenesis of NAFLD in the context of sleep and circadian abnormalities was explored. The relationship between various sleep disorders (obstructive briefs apnea, circadian rhythm disturbances, insufficient sleep) and NAFLD was analyzed. Effective non-pharmacological therapeutic options, such as lifestyle modification through diet and proper exercise regimen, were discussed. Promising options for pharmacological therapy are melatonin, vitamin E, thiazolidinediones, and fecal microbiota transplantation. According to Lin et al. [36], nocturnal hypoxia in patients with obstructive sleep apnea might be a risk factor in the progression of NAFLD. The intermittent hypoxia seen in obstructive sleep apnea may contribute to fibrotic changes in the liver [35].

Melatonin is important, not only as a chronobiotic but also playing a role as an antioxidant. The numerous actions of melatonin as a direct and indirect antioxidant were discussed by Reiter et al. in 2003 [37]. Melatonin acts as an antioxidant by directly scavenging free radicals, stimulating antioxidant enzymes, increasing the efficacy of mitochondrial oxidative phosphorylation, and reducing electron leakage, increasing the efficacy of other antioxidants. The mechanisms by which melatonin affects oxidative stress have been studied at the cellular and tissue level (liver, brain, kidney) [38,39]. In laboratory conditions, a gastro- and hepatoprotective effect of exogenous melatonin has been established [40,41]. The results suggest that melatonin ameliorates carbon tetrachloride-induced hepatic fibrogenesis in rats via inhibition of oxidative stress and proinflamatory cytokines production [41]. The rats injected subcutaneously with carbon tetrachloride for 6 weeks resulted in hepatic fibrotic changes, increased hydroxyproline and malondialdehyde (MDA) levels, and decreased glutathione peroxidase and superoxide dismutase levels, whereas melatonin reversed these effects. Melatonin inhibited the expression of nuclear factor-kappa B in liver tissue and decreased the production of proinflammatory cytokines, such as tumor necrosis factor-α (TNF-α) and interleucin-1ß (IL-1ß), from Kupffer cells in fibrotic rats. The effect of melatonin on the treatment of patients with NAFLD was evaluated in a study by Bahrami et al. [42]. This randomised double-blind, placebo-controlled study showed that administration of 6 mg/day melatonin for 12 weeks improved a number of factors associated with NAFLD, such as enzymes (aspartate aminotransferase, AST; alanine aminotransferase, ALT), anthropometric factors (weight, waist circumference, abdominal circumference), blood pressure, serum leptin levels, and the grade of fatty liver.

Other physiological effects of melatonin are related to its involvement in the processes of sexual maturation and reproduction [43] and in aging processes [27], possibly having an oncostatic action [27], enhancing the immune response [44].

Apart from the pineal gland, melatonin secretion has been found in extrapineal tissues. Based on a data from 326 literary sources, Acuña-Castroviejo et al. [45] presented data on melatonin formation in the retina, immune system, reproductive and gastrointestinal tract (enterochromaffin cells of the gastrointestinal mucosa, liver hepatocytes), and in the bile, saliva, and cerebrospinal fluid. From the reviewed data, it is assumed that melatonin has a double role in the body: synchronizing the organism’s functions and protecting the cells from oxidative/inflammatory damage.

Melatonin is mainly metabolized in the liver and, to a lesser extent, in the kidney [46,47], and, therefore, severe liver or kidney disease can affect the melatonin rhythm [48].

## 3. Pathogenetic Mechanisms of NAFLD

The processes underlying NAFLD pathogenesis remain unknown [49]. It is known that the pathogenetic mechanisms of hepatic steatosis are generally related to either increased delivery of fatty acids to the liver or defects in the complex process of very-low-density lipoprotein (VLDL) synthesis and export of triglycerides [3]. However, they do not answer the question why, in some patients, NAFLD progresses to inflammation, fibrosis, and cirrhosis and, in others, it does not. They also cannot explain what the leading factors are for the appearance of a more aggressive phenotype of NAFLD in some patients.

### 3.1. Hypothesis to Explain the Pathogenesis of the NAFLD

For many years, the initially accepted explanation for NAFLD progression was the two-hit hypothesis [50]. The first hit is associated with obesity and insulin resistance, which triggers hepatic de novo lipogenesis and impaired fatty acid transport and so increases the accumulation of fat in the liver (steatosis) [50,51]. These changes are not sufficient to cause inflammation and necrosis of the liver [52], but they increase susceptibility to risk factors (second hit) that lead to the progression of NAFLD to NASH, with subsequent development of cirrhosis and hepatocellular carcinoma [51]. Risk factors that contribute to the progression of NAFLD to NASH are, for example, endoplasmatic reticulum stress, perturbation of autophagy, mitochondrial dysfunction, and hepatocellular apoptosis, as well as an increase in inflammatory responces. The second hit involves a source of free radicals capable of inducing oxidative stress, which initiates significant lipid peroxidation, inflammation, and fibrosis [53].

It is considered that this concept provides a rationale for both the treatment and prevention of disease progression in steatosis of alcoholic and nonalcoholic causes [53] but cannot explain the several molecular and metabolic changes that take place in NAFLD [50]. It is believed that the most accepted hypothesis is the later multiple-hits hypothesis, with metabolic syndrome playing a major role due to insulin resistance and the proinflammatory process mediated by different proteins and immune components [2]. The identities of the multiple hits are different in each patient. The hypothesis was proposed by Tilg and Moschen [54], according to which hepatic inflammation in NASH is the result of many parallel-acting hits (multiple parallel-hits hypothesis). The latter includes a number of extrahepatic (pro-inflammatory gut microbes, addipose tissue insulin resistance) and intrahepatic stressors (oxidative stress of the liver, stress of the endoplasmic reticulum) [49,52]. About ten years later, Tilg, Adolph, and Moschen [55] made a critical review of this hypothesis, according to which, in addition to lipotoxicity of adipose tissue and changes in gut microbial functions, other “hits” with anti-inflammatory potential (e.g., dietary, genetic) are also likely to contribute to the inflammation and fibrosis in NAFLD. The role of various cellular factors in adipose tissue inflammation (cytokines, adipokines, chemokines) in obesity-related diseases is discussed. The key role of inflammation in the pathophysiology of NAFLD has been reported, but attention is drawn to the fact that large randomized controlled trials specifically aimed at elucidating the pathways leading to the inflammatory process are still lacking. The authors reasoned that a better understanding of this concept would lead to better clinical interpretation and the development of new therapeutic strategies for this endemic disease.

To explain the pathogenesis of NAFLD, a distinct-hit hypothesis has been proposed, according to which simple steatosis and NASH are two distinct entities with different pathogenetic pathways [56].

### 3.2. Experimental Data on the Role of Melatonin in the Pathogenesis of NAFLD

A number of experimental studies have studied the role of melatonin in different parts of the pathogenesis of NAFLD. In a comprehensive review from 2023, Ghosh et al. [49] disscussed the various potential factors that play a role in the pathogenesis of NAFLD (for example, high-fat diet, adipose tissue dysfunction, intrahepatic de novo lipogenesis, hepatic fat accumulation, insulin resistance, hepatic inflammation and inflammasome activation, mitochondrial dysfunction, oxidative stress, etc.) and the possible ameliorative effects of melatonin. Data from a number of studies support the hepatoprotective effect of melatonin in NAFLD. Obydah et al. [57] studied the possible role of melatonin, glutamine, and L-arginine in the prevention or attenuation of NAFLD in rats fed a high-fat, high-carbohydrate diet for six weeks. The obtained data confirm the role of carbohydrate (serum glucose) and lipid (triglycerides, total cholesterol, low-density lipoproteins) disorders in the development of steatosis. High liver enzymes (AST, ALT), low reduced glutathione, and high MDA (a marker of lipid peroxidation) indicate the essential role of oxidative stress in the pathogenesis of NAFLD. According to the study, the administration of melatonin 5 mg/kg/day in NAFLD rats has hypoglycemic and hypolipidemic effects, the decrease in liver enzymes and MDA, and the increase in reduced glutathione can be explained by the antioxidant properties of melatonin. Miguel et al. [58] also found reductions in steatosis, inflammation, and balloonization in NASH mice treated for 2 weeks with melatonin (20 mg/kg body weight) in comparison with a NASH group untreated with melatonin. The treatment with melatonin decreased lipoperoxidation (MDA), increased the activity of the superoxide dismutase and glutathione peroxidase, and reduced catalase activity in the liver, indicating its antioxidant effects. Damage index as the biomarker of DNA damage is significantly lower in the NASH group treated with melatonin compared to the NASH group untreated with melatonin, both in blood and in the liver. Data from the study of Soriano et al. [59], however, do not confirm the beneficial effect of melatonin on hepatic steatosis.

A review of data from experimental studies indicated that melatonin exerts its effects on NAFLD at the cellular, subcellular, and molecular levels. Different types of liver cells are involved in the progression of the disease—hepatocytes, macrophages (Kupffer cells and monocyte-derived macrophages), stellate cells, and many others [60,61,62,63]. In a NASH mouse model, melatonin directly attenuated the lipid storage in the primary hepatocytes, the inflammation by supressing activation of macrophages, and also reducing liver fibrosis by attenuating hepatic stellate cells [60].

It is known that as a result of extracellular signals, a number of signaling pathways are activated in cells, which cause immediate or long-term responses [64]. Intracellular signaling pathways include mitogen-activated protein (MAP) kinases, which are activated by phosphorylation cascades and are important in the control of cell differentiation and proliferation and cell death [64,65,66]. Three main groups of MAP kinases have been identified in mammalian cells: extracellular signal-regulated kinases, c-Jun N-terminal kinases, and p38 MAP kinases [64]. Substrates for the action of MAP kinases are membrane (phospholipase A2), cytoplasmic (e.g., cytoskeletal proteins), and nuclear (transcription factors) proteins [65]. Melatonin likely decreases inflammation via the MAPK-JNK/P38 signaling pathway [67]. Melatonin treatment significantly reduced the mRNA level of pro-inflammatory cytokines (TNF-α, IL-1β, interleukin-6 (IL-6)), which were increased in high-fat-diet (HFD)-fed mice compared with regular-diet-fed mice. The expression of the key mediators in cellular responces to extracellular stimuli total P38 and total c-Jun N-terminal kinases was similar in these two groups, but their phosphorylated forms, which were increased in high-fat-diet-fed mice, were reduced by melatonin administration.

Another way for melatonin to intervene in the pathogenesis of NAFLD is by inhibiting the nuclear orphan receptor subfamily 4 group A member 1 (NR4A1) in hepatocytes [68]. Diet-induced NAFLD in mice was found to increase NR4A1 expression in hepatocytes, leading to the activation of deoxyribonucleic acid (DNA)-dependent protein kinase catalytic subunit (DNA-PKcs) and p53. Melatonin supplementation inhibits NR4A1, which, in turn, attenuates hepatic lipogenesis and fibrosis, blocks mitochondrial fission, restores levels of antioxidant factors (superoxide dismutase, glutathione), alleviates the mitochondrial lipid oxidation (decreased MDA generation), and represses the p53 activation. Melatonin protects mitochondrial respiratory function via improving mitophagy. Thus, by inhibiting the NR4A1/DNA-PKcs/p53 pathway, melatonin improves mitochondrial function and largely reverses the pathogenesis of HFD-induced NAFLD.

NLRP3 (NLR family pyrin domain containing 3) inflammasome is an intracellular multimeric protein complex that activates caspase-1 and proinflammatory cytokines IL-1β and interleukin 18 (IL-18) [69]. Factors, such as mitochondrial dysfunction and generation of reactive oxygen species (ROS), are hypothesized to activate NLRP3 inflammasomes [70]. In diabetes mellitus, the high concentration of glucose is related to auto-oxidation of glucose to form free radicals [71]. The increased formation of free radicals in diabetes can be due to mitochondrial (respiratory chain, non-enzymatic source of free radicals) and non-mitochondrial sources (lipoxygenase, xanthine oxidase, P450 enzymes). Yu et al. [72] studied the action of melatonin toward the development of diabetes-related NAFLD and analyzed the underlying mechanisms of this process by inhibiting NLRP3 inflammasome activation. In db/db mice, melatonin treatment for 8 weeks significantly reduced metabolic parameters in blood, attenuate hepatic steatosis, ballooning injury, and lobular inflammation, decreasing the level of oxidative stress and improving mitohondrial function by upregulating mitocchondrial membrane potential. These beneficial effects of melatonin on NAFLD relate to the significantly reduced expression of NLRP3 inflammasome-related mRNA and proteins, caspase-1, IL-1β, and IL-18 in the liver tissue of diabetic mice. These data suggest that melatonin may protect against diabetes-related NAFLD by supressing the activation of the NLRP3 inflammasome. More detailed information on the mechanism by which melatonin exerts a suppressive effect on NLRP3 inflammasomes in the HFD-induced murine NASH model was provided by the study of Saha et al. [73]. According to this study, melatonin (10, 20 mg/kg) improves inflammation and other pathophysiological changes in HFD-induced murine NASH via the inactivation of NLRP3 inflammasomes via the suppression of toll-like receptor 4 (TLR4)/nuclear factor-kappa B (NF-kB) pathway and modulation of adenosine triphosphate (ATP)-dependent P2X7 receptor. Saha et al. [73] hypothesized that P2X7-mediated activation of NLRP3 may be the third hit among multiple hits that are relevant to the progression of NAFLD. Detailed information on the relationship between TLRs, NF-kB, and P2X7 receptor and regulation of inflammatory responses is presented in a number of literature reviews [74,75,76]. From these, it can be summarized that as transmembrane proteins, TLRs have N-terminal extracellular repeats that recognize specific components of the pathogen, which is a key factor in the induction of inflammatory responses [74]. Signaling cascades are activated and lead to the activation of NF-kB. After activation, NF-kB controls the transcription of genes related to the formation of cytokines (IL-6, IL-8, TNFα), chemokines (IL-18), adhesion molecules (VCAM-1, ICAM-1), and anti-apoptotic factors (caspase) [75]. In addition, NF-kB is involved in the regulation of NLRP3 inflammasomes. He believes that the P2X7 receptor, a non-specific cation channel, plays an important role in inflammatory processes, the activation of which initiates signaling cascades associated with the release of caspase, cytokines, and some MAPKs (P38MAPK, JNKs) [76].

An important regulator of carbohydrate and lipid metabolism in the liver is the silent information regulator 1 (SIRT1), a NAD+-dependent protein deacetylase, whose activity is influenced by nutritional, hormonal, and environmental signals [77]. Conditions, such as starvation and physical exertion, increase the concentration of NAD+ in cells, which stimulates SIRT1 activity. SIRT1 is involved in the regulation of local and systematic metabolic homeostasis and pancreatic insulin secretion, and it is an essential molecular link between nutrients, inflammation, and metabolic dysfunction. SIRT1 is expressed in the liver, brain, adipose tissue, heart, and molecular targets for its action on histones, tumor-suppressor protein p53, NF-κB, etc. [78]. According to Ren et al. [79], melatonin restores impaired autophagy in fat diet/chronic intermittent hypoxia-induced liver injury in a mouse model by activating SIRT1 signaling. The expression and activity of SIRT1 can be modulated by more than 16 microRNAs; among these is miR-34a [80]. MicroRNAs are small non-protein coding RNAs, which are involved in gene regulation [81]. It is considered that microRNAs are one of the significant diagnostic and therapeutic targets in NAFLD [82]. It has been found that oral melatonin (10 mg/kg) directly decreases the hepatic miR-34a-5p expression levels, resulting in the upregulation of its target SIRT1 in dietary obese mice, only in the presence of full SIRT1 availability [83]. Thus, melatonin attenuated steatosis and lipid peroxidation in the liver, reduced endoplasmatic reticulum stress and lipogenesis, and recovered autophagic flux.

It is assumed that one of the risk factors for the onset and progression of NAFLD is air pollution, and the attention of researchers is directed to fine dust particles with a diameter of less than 2.5 µm (PM_2.5_) [84]. A study by Du et al. [85] showed that in apoE^−/−^ mice, melatonin 20 mg/kg/day orally for 4 weeks alleviated PM_2.5_-triggered hepatic steatosis and liver damage by regulating ROS-mediated protein tyrosine phosphatase1B (PTP1B) and nuclear factor erythroid 2-related factor (Nrf2) signaling pathways. Exposure to PM_2.5_ resulted in liver steatosis (large lipid droplets, enlarged adipocytes, fatty degeneration) and oxidative stress (significantly increased values of MDA and 4-hydroxynonenal, increased activity of superoxide dismutase and glutathione peroxidase), which are alleviated by melatonin. According to the study, the accumulation of lipids is the result of the upregulation of PTP1B and the inhibition of Nrf2 signaling by P_2.5,_ inducing the overproduction of ROS, and with its antioxidant properties, melatonin suppresses the generation of reactive species and, thus, reduces the harmful consequences caused by the accumulation of P_2.5_.

The signaling pathways affected by melatonin in NAFLD and the effects of melatonin are summarized in Table 2.

## 4. Therapeutic Potentials of Melatonin in NAFLD

In 2019, Baiocchi et al. [86] expressed the opinion that all studies related to the potential therapeutic effects of melatonin on NASH in rodents and humans did not pinpoint the possible molecular mechanisms by which melatonin protects against NASH but, rather, focused only on the general antioxidant and cytoprotective properties of melatonin in this setting. Two years earlier, Zang et al. [87] analyzed the protective effect of melatonin on liver injuries induced by various factors and liver diseases, such as liver steatosis, non-alcohol fatty liver, hepatitis, liver fibrosis, liver cirrhosis, and hepatocarcinoma. Our review of experimental data on the role of melatonin in the pathogenesis of NAFLD shows that melatonin affects a number of signaling pathways, resulting in improved inflammation, oxidative stress, lipid and fat metabolism, and improved mitochondrial physiology. According to Sato et al. [88], melatonin has potential for novel treatments of liver diseases by decreasing oxidative stress or restoring circadian rhythms and functions. However, related studies of melatonin applied to clinical treatment for liver injuries and diseases are limited [87]. Mohammadi et al. [89] administered melatonin (10 mg/day), metformin (500 mg/day), and vitamin E (800 IU/day) to patients with NAFLD for six months and found that melatonin reduced serum aminotransferases, triglycerides, cholesterol, and fasting glucose when comparing these parameters before and after medication with melatonin. When comparing these indicators against a control group (received plasibo), only low-density lipoprotein and AST had significant changes. Based on the improvements shown via ultrasonography, the greatest improvement was demonstrated with metformin, and the authors concluded that metformin is a better choice for the treatment of these patients. Mohammadi et al. [89] suggested that melatonin can be considered an effective treatment of NAFLD, as this drug made improvements in different aspects of NAFLD injuries. Gonciarz et al. [90] evaluated the effects of 24 weeks of lifestyle intervention combined with 10 mg/day melatonin treatment (5 mg at 09:00 h and 5 mg at 21:00 h) on plasma liver enzyme levels of AST, ALT, gamma-glutamyl transpeptidase (GGT), alkaline phosphatase (ALP), concentrations of lipids (total cholesterol, triglycerides), glucose, and melatonin in 30 patients with NASH. A control group of 12 patients with NASH who received placebo was used for comparison. The study demonstrates that AST and GGT levels decreased significantly only in the melatonin-treated group. The decrease in median plasma ALT level in the melatonin-treated group at weeks 18 and 24 was significantly more intense (*p* < 0.5) than that observed in the control group; however, at follow-up, the difference between the two groups was not significant. The higher ALT, AST, and GGT levels shown at follow-up in comparison with those found at the 18th and 24th weeks of treatment reflected the high efficacy of melatonin, linked closely to the period of medicine administration. Plasma concentration of melatonin (pg/mL) in the melatonin-treated group averaged 7.5 ± 3.5 at baseline and increased to 52.5 ± 17.5 at the 24th week, no patients complained of somnolence, and no significant side-effects were observed. Cichoz-Lach et al. [91] evaluated the effects of melatonin and L-tryptophan on selected biochemical parameters and proinflammatory cytokines of blood in 45 patients with NASH divided into three groups: the first group received preparation Essentiale forte three times a day and L-tryptophan 500 mg twice a day; the second group received Essentiale forte in the above doses and melatonin 5 mg twice a day; the third group received only Essentiale forte three times a day. The treatment lasted 4 weeks. In all participants, plasma biochemical parameters (ALT, AST, ALP, GGT, bilirubin, total cholesterol, triglycerides, LDL, HDL) and cytokines (IL-1, IL-6, and TNF-α) were measured after 4 weeks of treatment and were compared with the results evaluated at the start of the study. The study showed that the addition of melatonin or its precursor, L-tryptophan, to Essentiale forte therapy resulted in a statistically significant decrease in the plasma levels of key pro-inflammatory cytokines, such as IL-1, IL-6, and TNF-α. This effect can be explained by the antioxidant action of melatonin and leads to an improvement in the therapy of NASH. The beneficial effect is also accompanied by a decrease in GGT and triglyceride levels. Based on the results obtained, these researchers suggested that treatment with melatonin is very important in the prevention of the progression of liver damage in NAFLD and NASH. A similar design was used in the study of Celinski et al. [92], which also determined the effects of tryptophan and melatonin on the biochemical parameters in patients with NAFLD. In addition, they evaluated the effects of tryptophan and melatonin in improvements of liver tissue in selected NAFLD patients (*n* = 9) after 14 months of a treatment period. Significantly reduced activity of GGT and values of triglycerides, LDL-cholesterol, IL-1, IL-6, and TNF-α were found in the groups that received melatonin and tryptophan compared to the group that received only Essentiale forte. The study findings demonstrated that melatonin and tryptophan substantially reduce the levels of pro-inflammatory cytokines and improve some parameters of fat metabolism in patients with NAFLD. In a few patients with NASH, melatonin and tryptophan reduced the inflammation in the liver. It was concluded that melatonin is worth considering for the therapy of NAFLD, especially in patients with impaired fat metabolism (hypertriglyceridemia and hyper-LDL cholesterolemia). No side effects of melatonin and tryptophan were observed; for instance, no patients complained of excessive sleepiness and/or dizziness. Pakravan et al. [93] studied the effect of melatonin in 100 patients with NAFLD aged 22 to 65 years, divided into two groups: a case group (*n* = 50) who received melatonin tablets twice a day for 6 weeks, and a control group (*n* = 50) who received a placebo twice daily for the same period of time. During the study, the patients followed the same diet and exercise regime. Results showed that in the case group, the mean of weight, waist, systolic and diastolic blood pressure, high-sensitive C-reactive protein, and ALT after treatment was significantly decreased compared to baseline; also, melatonin significantly decreased diastolic blood pressure, AST, and high-sensitive C-reactive protein in case group more than the control group. In addition, most of the patients who received melatonin grade of fatty liver improved more than the controls. These results demonstrated that the use of melatonin in patients with NAFLD was more affected than placebo, with no serious side effects. Melatonin significantly decreases liver enzymes in cases more than the placebo; therefore, the use of melatonin in patients with NAFLD can be effective.

It is hypothesized that new compounds that act as specific melatonin agonists or antagonists will contribute to a better understanding of melatonin’s mechanism of action [29]. Melatonin analogues (agonists and antagonists) differ in their chemical structure and affinity for melatonin receptors [94]. Currently, powerful, lipophilic, non-selective MT1/MT2 high-exposure agonists in the brain, such as Ramelteon, Agomelatine, Tazimelteon, and prolonged-release melatonin (Circadin), are approved for the treatment of insomnia, depression, and circadian rhythm sleep–wake disorders [95]. In a systematic review, Freiesleben and Furczyk [96] evaluated the potential risk posed by agomelatine as an antidepressant in inducing liver injury. Agomelatine was found to be associated with higher rates of liver injury than both placebo and the four active comparator antidepressants used in the clinical trials for agomelatine, with rates as high as 4.6% for agomelatine compared to 2.1% for placebo, 1.4% for escitalopram, 0.6% for paroxetine, 0.4% for fluoxetine, and 0% for sertraline. The review also provided evidence for the existence of a positive relationship between agomelatine dose and liver injury. According to researchers, it is essential that clinicians continue to monitor liver function frequently, as prescribed by the manufacturer of agomelatine. Early detection followed by best-practice treatment plan reactions (e.g., treatment discontinuation) remain the most efficient responses toward possible manifestations of liver damage. Ferreira et al. [97] reported the discovery of a new powerful melatonin receptor agonist, benzoimidazole derivative compound 10b, which reduced weight gain, liver triglycerides, and steatosis in HFD rats. Two-month oral administration of 10b in high-fat-diet rats led to a reduction in body weight gain, with superior results on hepatic steatosis and triglyceride levels. An early toxicological assessment indicated that 10b (also codified as ACH-000143) was devoid of genotoxicity, and there were behavioral alterations at doses up to 100 mg/kg p.o. Based on its efficacy, oral pharmacokinetics, and safety, compound 10b was selected for further investigation as a candidate drug against NAFLD/NASH.

### Melatonin Side Effects

In 2001, Chung [98] reported that for 6 weeks, three patients attended the emergency department after attempting suicide by taking an overdose of melatonin. Their hospital stay was uneventful, but the report states that the emergency physicians were still unfamiliar with the management of melatonin “overdose”, and it is advisable to monitor for adverse effects, such as drowsiness, confusion, tachycardia, and hypothermia. In 2005, Waldron et al. [99] drew attention to the fact that there was a shortage of randomized controlled trials to demonstrate the efficacy of melatonin therapy, and that the lack of pharmacokinetics, pharmacodynamics, and toxicology data limits knowledge of therapeutic dose ranges, formulations, and adverse effects. Later, in 2017, Erland and Saxena [100] quantified melatonin in 30 commercial supplements, comprising different brands and forms, and screened supplements for the presence of serotonin. The melatonin content was found to range from −83% to +478% of the labelled content, and serotonin was identified in eight of the supplements at levels of 1 to 75 μg. The significant variability in the melatonin content of analyzed additives and the presence of serotonin indicate the pressing need for mechanisms to monitor the melatonin content in these products, which will ensure the safety of supplements. In a number of countries (United Kingdom, Japan, Australia, European Union, and, most recently, Canada), exogenous melatonin is regarded as a medicine and available only through prescription [101].

Buscemi et al. [102] conducted a systematic review of the efficacy and safety of exogenous melatonin in managing secondary sleep disorders and sleep disorders accompanying sleep restriction, such as jet lag and shift-work disorder. The most commonly reported adverse events were headaches, dizziness, nausea, and drowsiness, but the occurrence of these outcomes was similar for melatonin and placebo. Lemoine et al. [103] investigated the efficacy, safety, and withdrawal phenomena associated with 6–12 months prolonged-release melatonin treatment in 244 adults with primary insomnia. In 7% of the patients, the adverse events were considered by the investigator to be definitely, probably, or possibly related to the study medication. Of these, the most commonly reported adverse events were dizziness in four patients (1.6%) and headache in three patients (1.2%). No noticeable changes were found in hematologic and biochemical laboratory tests at any timepoint during the study. Khezri and Merate [104] evaluated the effects of melatonin premedication on anxiety and pain scores of patients, operating conditions, and intraocular pressure during cataract surgery under topical anesthesia. Sixty patients were randomly assigned to receive either sublingual melatonin 3 mg or placebo 60 min before surgery. Only one patient in the melatonin group complained of mild headache. Ismail and Mowafi [105] evaluated the effects of melatonin premedication on pain, anxiety, intraocular pressure, and operative conditions during cataract surgery under topical analgesia. Forty patients undergoing cataract surgery under topical anesthesia were randomly assigned into two groups (twenty patients each) to receive either a melatonin 10 mg tablet (melatonin group) or placebo tablet (control group) as oral premedication 90 min before surgery. One patient in the melatonin group complained of dizziness, and another patient in the control group suffered nausea. In a study by Esmat and Kassim [106], 75 patients were randomly divided into three groups: C group (*n* = 25), each patient received transdermal placebo patch, TDF group (*n* = 25), each patient received transdermal therapeutic system-fentanyl 50 μg/h, and TDM group (*n* = 25), each patient received transdermal therapeutic system containing 7 mg of melatonin. All patches were placed 2 h preoperatively and were applied to the skin in the subclavicular area. The patch was removed 12 h postoperatively. As regards side effects in this study, all cases of the three groups were hemodynamically stable, no patient developed hypoxia, and there were no reported intraoperative complications interfering with the course of surgery or interrupting the surgeons. Two patients in the C group suffered from nausea (*p* = 0.08). Regarding adverse effects in patients who received TDM, patients were more sedated (*p* < 0.05) and two patients were dizzy (*p* = 0.08). Baradari et al. [107] investigated the effect of preoperative oral melatonin on the severity of postoperative pain after lumbar laminectomy/discectomy; 80 patients were selected and randomly assigned into one of four groups. Patients in groups A, B, C, and D received 3, 5, and 10 mg melatonin or placebo tablets one hour before surgery, respectively. Two patients in group A, one patient in group B, and two patients in the placebo group had postoperative vomiting, but the difference between the groups in terms of postoperative vomiting was not statistically significant (*p* = 0.524).

## 5. Dietary Sources of Melatonin

Melatonin is an evolutionally very old molecule, which is synthesized in many organisms, such as bacteria, protists, fungi, macroalgae, plants, and animals [108]. Dietary sources of melatonin in animals and plant, the benefits of consuming melatonin-containing foods, and guidance on regulating dietary supplements of melatonin were presented in a review article by Meng et al. [109]. According to Binici et al. [110], there are only a limited number of studies examining the melatonin profiles in the human body, especially the effect of consumption of foods on melatonin profiles in the human body. Sae-Teaw et al. [111] tested whether the consumption of fruits or fruit juice containing melatonin would influence the serum melatonin concentration and antioxidant status. Thus, 12 healthy male volunteers took either juice extracted from one kilogram of orange or pineapple or two whole bananas, with a 1 wk washout period between the fruit or fruit juices. The highest serum melatonin concentration was observed at 120 min after fruit consumption, and compared with before consumption levels, their values were significantly increased for pineapple (146 vs. 48 pg/mL, *p* = 0.002), orange (151 vs. 40 pg/mL, *p* = 0.005), and banana (140 vs. 32 pg/mL, *p* = 0.008), respectively. Significantly increased serum antioxidant capacity following fruit consumption was determined through a ferric-reducing antioxidant power assay (7–14% increase, *p* ≤ 0.004) and oxygen radical antioxidant capacity assay (6–9% increase, *p* = 0.002). The study showed that tropical fruit consumption increases serum melatonin concentrations and raises the serum antioxidant capacity in healthy volunteers. According to Marhuenda et al. [112], melatonin concentration varies from pictograms/mL to nanograms/mL in fermented beverages, such as wine and beer, depending on the fermentation process. These low quantities, within a dietary intake, are enough to reach significant plasma concentrations of melatonin and are, thus, able to exert beneficial effects. The health-promoting effects arising from Mediterranean dietary habits have been attributed to the large intake of plant foodstuffs rich in bioactive phytochemicals, such as melatonin [113]. Melatonin present in plant foods may promote health benefits by virtue of its biological activities, and it may counteract pathological conditions related to carcinogenesis, diabetes, cardiovascular diseases, neurological disorders, and aging. It is likely that the health benefits attributed to a food or drink do not depend on only one compound present in it (phenolic, carotenoid, or other), but the combination of phytochemicals has been shown to improve their bioactivity through additive and synergistic effects.

## 6. Conclusions

The currently existing hypotheses about the pathogenesis of NAFLD do not provide a complete picture of the mechanisms underlying the disease and the causes of its progression. A review of experimental study data on the role of melatonin in the pathogenesis of NAFLD shows that melatonin executes its effects by influencing various signaling pathways. Melatonin administration improves mitochondrial function, reduces inflammatory markers and the formation of reactive oxygen species, restores levels of antioxidant factors, and improves liver steatosis and fibrosis in high-fat-diet-induced NAFLD and NASH. Clinical studies have shown that exogenous melatonin lowers hepatic aminotransferases and gamma-glutamyl transpeptidase and lipid profile indicators, and it is assumed that melatonin should be practically important in the prevention of the progression of liver damage in NAFLD and NASH. In addition to exogenous melatonin, melatonin agonists are used in clinical practice (Ramelteon, Agomelatine, Tazimelteon, prolonged-release melatonin Circadian), approved for the treatment of insomnia, depression, and disturbances in the sleep–wake circadian rhythms. In connection with evidence of liver function injury when using agomelaton, frequent monitoring of liver function is recommended. The use of melatonin receptor agonist ACH-000143 for the treatment of NAFLD is under investigation. In a number of surgical interventions, melatonin has been used for premedication. As side effects of melatonin administration, single cases of headache, dizziness, and nausea have been reported. The diversity of doses and routes of administration of exogenous melatonin in different studies are noteworthy. Melatonin is found in a number of foods (vegetables, fruit, nuts, meat) in different concentrations. There is evidence to show that eating foods rich in melatonin increases serum melatonin, but it is not yet clear how dietary melatonin consumption affects the melatonin profile in humans.

## Figures and Tables

**Table 1 biomedicines-11-01722-t001:** Metabolic and chronobiological actions of melatonin [30].

Metabolic Action	Chronobiological Action
Regulates energy flow to and from storages	Circadian synchronization of white adipose tissue metabolism
Regulates energy expenditure	Circadian synchronization of skeletal muscle metabolism
Potentiates central and peripheral actions of insulin	Circadian synchronization of insulin synthesis, secretion and action
Regulates glycemia and lipidemia	Circadian synchronization of hepatic metabolism

**Table 2 biomedicines-11-01722-t002:** Signaling pathways affected by mealtonin in NAFLD and effects of melatonin.

Autors/References	Model	Melatonin Dose/Treatment Duration	Signaling Pathway That Melatonin Affects	Melatonin Effects
Sun et al. [67]	Diet-induced NAFLD mice	10 mg/kg/day/ 12 weeks	MAPK-JNK/P38	Reduce inflammation (↓mRNA level of TNF-α, IL-1ß, IL-6), decreased body weight and liver weight
Zhou et al. [68]	Diet-induced NAFLD mice	20 mg/kg/day/ 12 weeks	NR4A1/DNA-PKcs/p53	Protects mitochondrial structure and function (blocked mitochondrial fission, reversed the reduction of mitophagy, ↓oxidative stress and calcium overload)
Yu et al. [72]	db/db mice	30 mg/kg/day/8 weeks	NLRP3 inflammasome	↓serum glucose, ↓insulin, ↓lipids, ↓hepatic enzymes; ↓IL-1ß and IL-18; attenuated steatosis, baloonbing injury, and lobular inflammation; ↓oxidative stress
Saha et al. [73]	Diet-induced NASH mice	10, 20 mg/kg/ 1 month	TLR4/NF-kB and P2X7R	Anti-obesogenic effect; ↓serum glucose, ↓lipids, ↓TNF-α, ↓IL-6; ↓IL-1ß and IL-18 in hepatocyte; ↓ intracellular ROS
Ren et al. [79]	Diet-induced obesity mice	10 mg/kg/day/6 weeks	SIRT1	Protective effect on the liver via activation of autophagy
Stacchiotti et al. [83]	Diet-induced NAFLD mice	10 mg/kg/day	Micro-RNA-34a-5p/Sirt 1	Improved metabolism and lipid peroxidation in the liver, attenuate steatosis and endoplasmatic reticulum stress
Du et al. [85]	apoE^−/−^ mice	20 mg/kg/day	PTP1B/Nrf2	↓formation of ROS, improves hepatic lipid metabolism

Abbreviations: Down arrow (↓) indicate melatonin-induced decrease. MAPK: mitogen-activated protein kinase; JNK: Jun N-terminal kinases; P38: total P-38 and phospho-P38; NR4A1: nuclear orphan receptor subfamily 4 group A member 1; DNA-PKcs: DNA-dependent protein kinase catalytic subunit; p53: protein, transcription factor; NLRP3 inflammasome: nucleotide-binding oligomerization domain-like receptor family pyrin domain containing 3 inflammasome; TLR: toll-like receptor; NF-kB: niclear factor-kappa B; P2X7R: receptor, a non-specific cation channel; ROS: reactive oxygen species; SIRT1: silent information regulator 1; PTP1B: protein tyrosine phosphatase1B; Nrf2: nuclear factor erythroid 2-related factor.

## Data Availability

Not applicable.

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
