# Peer review of "Experimental Data on the Role of Melatonin in the Pathogenesis of Nonalcoholic Fatty Liver Disease"

_biomedicines, 2023, doi:10.3390/biomedicines11061722_

Round 1
Reviewer 1 Report
The paper by Dr. Dimitar Terziev reviewed the beneficial role of melatonin for NAFLD. Melatonin has some beneficial effects on liver however the underlying mechanisms is still poor understanding. The authors summarized their biological interaction and updated the studies. This review might shed light on understanding of melatonin in NAFLD but some of the methods need to be clarified.
1. Who is he? (line 120)
2. Please define abbreviation of ROR/RZR, DNA-PKcs and ATP.
3. Please make sure if the reference is only one because the authors mentioned “other studies” in line 242. (line 243)
Author Response
Point 1: Who is he? (line 120)
Response 1: On line 120 “he” was replaced by “melatonin”.
Point 2: Please define abbreviation of ROR/RZR, DNA-PKcs and ATP
Response 2:
- ROR/RZR – abbreviation of retinoid-related orphan receptor (ROR) and retinoid Z receptor (RZR)
- DNA-PKcs – abbreviation of deoxyribonucleic acid-dependent protein kinase catalytic subunit
- ATP – abbreviation of adenosine triphosphate
In the article before these abbreviations are already indicated the expanded names (line 129, line 274 , and line 302 respectively). In connection with defining the abbreviation DNA-PKcs, for a clearer understanding of the text, small additions have been made to the information provided regarding reference [54].
Point 3: Please make sure if the reference is only one because the authors mentioned “other studies” in line 242. (line 243)
Response 3: On line 242/243 one author is listed and in the text of the review "Data from other studies however, do not confirm the beneficial effect of melatonin on hepatic steatosis [45]" is corrected to "Data from study of Soriano et al. [45], however, do not confirm the beneficial effect of melatonin on hepatic steatosis."
Reviewer 2 Report
In this manuscript the authors insisted that influence of melatonin on patients with different risk factors and degree of metabolic involvement of the liver, especially NASH. This would lead to refinement of the individual therapeutic approach and development of preventive measures.
I have some comments.
1. This review has no figures or tables and is very difficult to understand. Please write more clearly from the reader's point of view.
2. The perspective that melatonin might slow the progression of NAFLD and NASH is very interesting. Do you have any clinically supportive evidence?
3. Do you have any information on foods or supplements that contain a lot of melatonin? Also, please state whether melatonin receptor agonists have the same NASH inhibitory effect.
4. Supplements containing melatonin are regulated in France. The safety of melatonin is also controversial. Please describe any issues that may arise for clinical use.
There is confusion in spelling errors and pronoun usage. Please check again.
Author Response
Point 1: This review has no figures or tables and is very difficult to understand. Please write more clearly from the reader's point of view.
Response 1: For ease of text perception, two tables have been added – one (Table 1) is in the section "Melatonin, general data, mechanisms of action and physiological significance", and the second (Table 2) is in the part "Experimental data on the role of melatonin in the pathogenesis of NAFLD"
Melatonin probably consolidate the whole rhythmic organisation by it’s chronobiotics properties [28, 29]. It is believed that through its prospective effects, melatonin can control the circadian time of the suprachiasmal nucleus, and pancreatic ß-cells and islets and the metabolically most important tissues (liver, muscle, and adipose tissue) are targets for melatonin [28]. As a strong chronobiotic, melatonin influences the circadian distribution of metabolic processes, synchronizing them to the activity-feeding/rest-fasting cycle [30]. Melatonin is an important player in regulation of energy balance and carbohydrate, and adipocyte metabolism [29, 30]. Table 1 summarizes the metabolic and chronobiological effects of melatonin that influence energy metabolism, and ultimately body weight.
Тable 1. Metabolic and chronobiological actions of melatonin [30]
Metabolic action |
Chronobiological action |
Regulates energy flow to and from storages |
Circadian synchronization of white adipose tissue metabolism |
Regulates energy expenditure |
Circadian synchronization of skeletal muscle metabolism |
Potentiates central and peripheral actions of insulin |
Circadian synchronization of insulin synthesis, secretion and action |
Regulates glycemia and lipidemia |
Circadian synchronization of hepatic metabolism |
Summary data on the signalling pathways affected by mealtonin in NAFLD and the effect of this influence are presented in Table 2.
Table 2. Signalling pathways affected by mealtonin in NAFLD and the effect of this influence
References |
Model |
Dose/treatment duration |
Signaling pathway that melatonin affects |
Effects of melatonin |
Sun et al. [53] |
Diet-induced NAFLD mice |
10 mg/kg/day/12 weeks |
MAPK-JNK/P38 |
Reduce inflammation (↓mRNA level of TNF-ɑ, IL-1ß, IL-6), decreased body weight and liver weight |
Zhou et al. [54] |
Diet-induced NAFLD mice |
20 mg/kg/day/12 weeks |
NR4A1/DNA-PKcs/p53 |
Protects mitochondrial structure and function (blocked mitochondrial fission, reversed the reduction of mitophagy, ↓oxidative stress and calcium overload) |
Yu et al. [58] |
db/db mice |
30 mg/kg/day/8 weeks |
NLRP3 inflammasome |
↓serum glucose, insulin, lipids, hepatic enzymes; IL-1ß and IL-18; attenuated steatosis, baloonbing injury, and lobular inflammation; ↓oxidative stress
|
Saha et al. [59] |
Diet-induced NASH mice |
10, 20 mg/kg/1 month |
NLRP3 inflammasome via TLR4/NF-kB and P2X7R |
Anti-obesogenic effect; ↓serum glucose, lipids, TNF-ɑ, IL-6;↓IL-1ß and IL-18 in hepatocyte; ↓ intracellular ROS |
Ren at al. [65] |
Diet-induced obesity mice |
10 mg/kg/day/6 weeks |
SIRT1 |
Protective effect on the liver via activation of autophagy |
Stacchiotti et al. [69] |
Diet-induced NAFL mice
|
10 mg/kg/day |
Micro-RNA-34a-5p/Sirt 1 |
Improved metabolism and lipid peroxidation in the liver, attenuate steatosis and endoplasmatic reticulum stress |
Du et al. [71] |
аpоЕ-/- mice |
20 mg/kg/day/4 weeks |
PTP1B/ Nrf2 |
↓formation of reactive oxygen species, improves hepatic lipid metabolism |
Abbreviations: Down arrow (↓) indicate melatonin-induced decrease. MAPK-mitogen-activated protein kinase; JNK: Jun N-terminal kinases; P38: total P-38 and phospho- P38; NR4A1: nuclear orphan receptor subfamily 4 group A member 1; DNA-PKcs: DNA-dependent protein kinase catalytic subunit; p53: protein, transcription factor; NLRP3 inflammasome: nucleotide-binding oligomerization domain-like receptor family pyrin domain containing 3 inflammasome; TLR: toll-like receptor; NF-kB: niclear factor-kappa B; P2X7R: receptor, a non-specific cation channel; SIRT1: Silent information regulator 1; PTP1B: protein tyrosine phosphatase1B ; Nrf2: nuclear factor erythroid 2-related factor
Point 2: The perspective that melatonin might slow the progression of NAFLD and NASH is very interesting. Do you have any clinically supportive evidence?
Response 2: In 2019, Baiocchi et al. [87] express the opinion that all studies related to the potential therapeutic effects of melatonin on NASH in rodents and humans did not pinpoint the possible molecular mechanisms by which melatonin protects against NASH, but rather, focused only on the general antioxidant and cytoprotective properties of melatonin in this setting. Two years earlier, Zang et al. [88] analyzed the protective effect of melatonin on liver injuries induced by various factors and liver diseases such as liver steatosis, non-alcohol fatty liver, hepatitis, liver fibrosis, liver cirrhosis, and hepatocarcinoma. Our review of experimental data on the role of melatonin in the pathogenesis of NAFLD shows that melatonin affects a number of signaling pathways, resulting in improved inflammation, oxidative stress, lipid and fat metabolism, improved mitochondrial physiology. Аccording to Sato et al. [89] melatonin has potentials for novel treatments of liver diseases by decreasing oxidative stress or restoring circadian rhythms and functions. However, related studies of melatonin applied to clinical treatment for liver injuries and diseases are limited [88]. Mohammadi et al. [90] administered melatonin (10 mg/day), metformin (500mg/day) and vitamin E (800IU/day) to patients with NAFLD for six months and found that melatonin reduced serum aminotransferases, triglycerides, cholesterol and fasting glucose when comparing these parameters before and after medication with melatonin. When comparing these indicators against a control group (received plasibo), only low density lipoprotein and AST had significant changes. Based on the improvements shown by ultrasonography, the greatest improvement was demonstrated by metformin and the autors concluds that metformin is a better chiose for the treatment of these patients. Mohammadi et al. [90] suggest that melatonin be considered an effective treatment of NAFLD as this drug made improvement in different aspects of NAFLD injuries. Gonciarz et al. [91] evaluated the effects of 24 weeks of lifestyle intervention combined with 10 mg/day melatonin treatment (5 mg at 09:00 h and 5 mg at 21:00 h) on plasma liver enzymes levels AST, ALT, gamma-glutamyl transpeptidase (GGT), alkaline phosphatase (ALP)], concentrations of lipids (total cholesterol, triglycerides), glucose and melatonin in 30 patients with NASH. A control group of 12 patients with NASH who received placebo was used for comparison. The study demonstrates that AST and GGT levels decrease significantly only in melatonin-treated group. The decrease in median plasma ALT level in melatonin-treated group at week 18 and week 24 was significantly more intense (P<0.5) than that observed in control group, however at follow-up the difference between the two groups was not significant. The higher ALT, AST and GGT levels shown at follow-up in comparison with that found at 18th and 24th week of treatment reflected the high efficacy of melatonin, linked closely to the period of medicine administration. Plasma concentration of melatonin (pg/ml) in melatonin-treated group averaged 7.5±3.5 at baseline and increased to 52.5±17.5 at 24th week, and no patients complained somnolence and no significant side-effects were observed. Cichoz-Lach et al. [92] evaluate the effects of melatonin and L-tryptophan on selected biochemical parameters and proinflammatory cytokines of blood in 45 patients with NASH divided into three groups: the first group received preparation Essentiale forte 3 times a day and L-tryptophan 500 mg twice a day; the second group received Essentiale forte in the above doses and melatonin 5 mg twice a day; the third group received only Essentiale forte three times a day. Treatment lasted 4 weeks. In all participants plasma biochemical parameters (ALT, AST, ALP, GGT, bilirubin, total cholesterol, triglycerides, LDL, HDL) and cytokines (IL-1, IL-6 and TNF-ɑ) were measured after 4 weeks of treatment and were compared with the results evaluated at the start of the study. The study showed that the addition of melatonin or its precursor, L-tryptophan to Essentiale forte therapy resulted in a statistically significant decrease in plasma levels of key pro-inflammatory cytokines such as IL-1, IL-6 and TNF-ɑ. This effect can be explained by the antioxidant action of melatonin and leads to an improvement in the therapy of NASH. The beneficial effect is also accompanied by a decrease in GGT and triglycerides levels. Based on the results obtained these researchers suggests that treatment with melatonin should be practically very important in prevention of the progression of liver damage in NAFLD and NASH. A similar design has the study of Celinski et al. [93] which also determine the effects of tryptophan and melatonin on the biochemical parameters in patients with NAFLD. In addition it is evaluate the effects of tryptophan and melatonin in improvement of liver tissue in selected NAFLD patients (n = 9) after 14-months of treatment period. Significant reduced activity of GGT and values of triglycerides, LDL-cholesterol, IL-1, IL-6 and TNF-ɑ were found in the groups received melatonin and tryptophan, compared to group received only Essentiale forte. The study findings demonstrate that melatonin and tryptophan substantially reduce the levels of pro-inflammatory cytokines and improve some parameters of fat metabolism in patients with NAFLD. In few patients with NASH melatonin and tryptophan reduced the inflammation in liver. It has been concluded that melatonin is worth considering for the therapy of NAFLD, especially in patients with impaired fat metabolism (hypertriglyceridemia and hyper-LDL cholesterolemia). No side effects of melatonin and tryptophan were observed; for instance none of patients complained on excessive sleepiness and/or dizzines. Pakravan et al. [94] studied the effect of melatonin in 100 patients with NAFLD aged 22 to 65 years, divided into two groups: a case group (n=50) received melatonin tablets twice a day for 6 weeks, and a control group (n=50) received twice daily placebo for the same period of time. During the study, the patients followed the same diet and exercise regime. Results showed that in case group the mean of weight, waist, systolic and diastolic blood pressure, high sensitive C-reactive protein and ALT after treatment was significantly decreased compare to baseline, also, melatonin significantly decrease diastolic blood pressure, AST, and high sensitive C-reactive protein in case group more than control group. In addition, in most of the patients who received melatonin grade of fatty liver improved than controls. These results demonstrated that use of melatonin in patients with NAFLD was more affected than placebo with no serious side effects. Melatonin significantly decrease liver enzymes in cases than placebo, therefore, the use of melatonin in patients with NAFLD can be effective.
Point 3: Do you have any information on foods or supplements that contain a lot of melatonin? Also, please state whether melatonin receptor agonists have the same NASH inhibitory effect.
Response 3:
- Dietary sources of melatonin
Melatonin is an evolutionally very old molecule, which is synthesized in many organisms such as bacteria, protists, fungi, macroalgae, plants and animals [109]. Dietary sources of melatonin in animal and plant, benefits of consuming melatonin-containing foods and guidance on regulating dietary supplement of melatonin are presented in review article of Meng et al [110]. According to Binici et al [111] there are only a limited number of studies examining the melatonin profiles in the human body, especially the effect of consumption of foods on melatonin profiles in the human body. Sae-Teaw et al. [112] tested whether the consumption of fruits or fruit juice containing melatonin would influence the serum melatonin concentration and antioxidant status. 12 healthy male volunteers took either juice extracted from one kilogram of orange or pineapple or two whole bananas, with a 1-wk washout period between the fruit or fruit juices. The highest serum melatonin concentration was observed at 120 min after fruit consumption, and compared with before consumption levels, their values were significantly increased for pineapple (146 versus 48 pg/mL P = 0.002), orange (151 versus 40 pg/mL, P = 0.005), and banana (140 versus 32 pg/mL, P = 0.008), respectively. Significantly increased serum antioxidant capacity following fruit consumption was found determined by ferric reducing antioxidant power assay (7–14% increase, P ≤ 0.004) and oxygen radical antioxidant capacity assay (6–9% increase, P = 0.002). The study shows that tropical fruit consumption increases serum melatonin concentrations and races the serum antioxidant capacity in healthy volunteers. According to Marhuenda et al. [113] melatonin concentration varies from pictograms/mL to nanograms/mL in fermented beverages such as wine and beer, depending on the fermentation process. These low quantities, within a dietary intake, are enough to reach significant plasma concentrations of melatonin, and are thus able to exert beneficial effects. The health-promoting effects arising from Mediterranean dietary habits have been attributed to the large intake of plant foodstuffs rich in bioactive phytochemicals, such as melatonin [114]. Melatonin present in plant foods may promote health benefits by virtue of its biological activities, and it may counteract pathological conditions related to carcinogenesis, diabetes, cardiovascular diseases, neurological disorders and ageing. It is likely that the health benefits attributed to a food or drink do not depend on only one compound present in it (phenolic, carotenoid or other), but the combination of phytochemicals has been shown to improve their bioactivity, through additive and synergistic effects.
- Whether melatonin receptor agonists have the same NASH inhibitory effect
It is hypothesized that new compounds that act as specific melatonin agonists or antagonists will contribute to a better understanding of melatonin's mechanism of action [29]. Melatonin analogues (agonists and antagonists) differ in their chemical structure and affinity for melatonin receptors [95]. Currently powerful, lipophilic, non-selective MT1/MT2 high exposure agonists in the brain, such as Ramelteon, Agomelatine, Tazimelteon, prolonged-release melatonin (Circadin) are approved for treatment of insomnia, depression and circadian rhythms sleep-wake disorders [96]. In a systematic review Freiesleben at Furczyk [97] evaluate the potential risk posed by agomelatine as an antidepressant of inducing liver injury. Agomelatine was found to be associated with higher rates of liver injury than both placebo and the four active comparator antidepressants used in the clinical trials for agomelatine, with rates as high as 4.6% for agomelatine compared to 2.1% for placebo, 1.4% for escitalopram, 0.6% for paroxetine, 0.4% for fluoxetine, and 0% for sertraline. The review also provides evidence for the existence of a positive relationship between agomelatine dose and liver injury. According to researchers it is essential that clinicians continue to monitor liver function frequently, as prescribed by the manufacturer of agomelatine. Early detection, followed by best practice treatment plan reactions (e.g., treatment discontinuation), remain the most efficient responses toward possible manifestations of liver damage. Ferreira et al. [98] reported the discovery of a new powerful melatonin receptor agonist, benzoimidazole derivative compound 10b, that reduced weight gain, liver triglycerides and steatosis in HFD rats. Two-month oral administration of 10b in high-fat diet rats led to a reduction in body weight gain with superior results on hepatic steatosis and triglyceride levels. An early toxicological assessment indicated that 10b (also codified as ACH-000143) was devoid of genotoxicity, and behavioral alterations at doses up to 100 mg/kg p.o. Based on its efficacy, oral pharmacokinetics and safety, compound 10b, are selected for further investigation as a candidate drug against NAFLD/NASH.
Point 4: Supplements containing melatonin are regulated in France. The safety of melatonin is also controversial. Please describe any issues that may arise for clinical use.
Response 4: Melatonin side effects
In 2001 Chung [99] reported that for 6 weeks 3 patients attended the emergency department after committing suicide by taking an overdose of melatonin. Their hospital stay was uneventful, but the report states that the emergency physicians were still unfamiliar with the management of melatonin "overdose" and it is advisable to monitor for adverse effects such as drowsiness, confusion, tachycardia and hypothermia. In 2005 Waldron et al. [100] draw attention to the fact that there is a shortage of randomized controlled trials to demonstrate the efficacy of melatonin therapy and that the lack of pharmacokinetics, pharmacodynamics, and toxicology data limits knowledge of therapeutic dose ranges, formulations, and adverse effects. Later, in 2017 Erland and Saxena [101] quantified melatonin in 30 commercial supplements, comprising different brands and forms and screened supplements for the presence of serotonin. The melatonin content was found to range from -83% to +478% of the labelled content and serotonin was identified in eight of the supplements at levels of 1to 75 μg. The significant variability in the melatonin content of ananlyzed additives and the presence of serotonin in them indicates the pressing need for mechanisms to monitor the melatonin content in the products, which will ensure the safety of supplements. In a number of countries (United Kingdom, Japan, Australia, European Union and most recently Canada) exogenous melatoninth is regarded as a medicine and available only by prescription [102].
Buscemi et al [103] conducted a systematic review of the efficacy and safety of exogenous melatonin in managing secondary sleep disorders and sleep disorders accompanying sleep restriction, such as jet lag and shiftwork disorder. The most commonly reported adverse events were headaches, dizziness, nausea, and drowsiness, but the occurrence of these outcomes was similar for melatonin and placebo. Lemoine et al. [104] investigate the efficacy, safety, and withdrawal phenomena associated with 6–12 months prolonged-release melatonin treatment in 244 adults with primary insomnia. In 7% of the patients, the adverse events were considered by the investigator to be definitely, probably, or possibly related to study medication. Of these, the most commonly reported adverse events were dizziness in four patients (1.6%) and headache in three patients (1.2%). No noticeable changes were found in hematologic and biochemical laboratory tests at any time-point during the study. Khezri and Merate [105] evaluate the effects of melatonin premedication on anxiety and pain scores of patients, operating conditions, and intraocular pressure during cataract surgery under topical anesthesia. Sixty patients were randomly assigned to receive either sublingual melatonin 3 mg or placebo 60 min before surgery. Only one patient in the melatonin group complained of mild headache. Ismail and Mowafi [106] evaluated the effects of melatonin premedication on pain, anxiety, intraocular pressure, and operative conditions during cataract surgery under topical analgesia. Forty patients undergoing cataract surgery under topical anesthesia were randomly assigned into two groups (20 patients each) to receive either melatonin 10 mg tablet (melatonin group) or placebo tablet (control group) as oral premedication 90 min before surgery. One patient in the melatonin group complained of dizziness, and another patient in the control group suffered nausea. In a study of Esmat and Kassim [107] 75 patients were randomly divided into 3groups: C group, (n=25) each patient received transdermal placebo patch, TDF group, (n =25) each patient received transdermal therapeutic system-fentanyl 50 μg/h and TDM group, (n= 25) each patient received transdermal therapeutic system containing 7 mg of melatonin. All patches were placed 2 h preoperatively and were applied to the skin in the subclavicular area. The patch was removed 12 h postoperatively. As regards side effects in this study, all cases of the 3 groups were hemodynamically stable, no patient developed hypoxia and there were no reported intraoperative complications interfering with the course of surgery or interrupting the surgeons. Two patients in the C group suffered from nausea (p = 0.08). Regarding adverse effects in patients who received TDM, patients were more sedated (P < 0.05) and two patients were dizzy (p= 0.08). Baradari et al [108] investigate the effect of preoperative oral melatonin on the severity of postoperative pain after lumbar laminectomy/discectomy. 80 patients were selected and randomly assigned into one of four groups. Patients in group A, B, C, and D received 3, 5 and 10 mg melatonin or placebo tablets one hour before surgery, respectively. 2 patients in group A, 1 patient in group B and 2 patients in the placebo group had postoperative vomiting, but the difference between the groups in terms of postoperative vomiting was not statistically significant (P=0.524).
Reviewer 3 Report
In this review, Terziev and Terzieva summarize the role of melatonin in NAFLD. Melatonin and circadian rhythms are a hot topic and timely, but this review lacks many basic and critical information and does not provide useful knowledge for readers.
· According to the title, the authors seems to focus on the roles of melatonin in pathogenesis of NAFLD. Melatonin regulates circadian rhythms, and disrupted circadian rhythms are associated with lipid metabolism, fat deposition, and diabetes. Melatonin supplementation could improve pathogenesis via improving circadian rhythms. There are some studies showing that, but this review does not mention about this at all.
· Reactive oxygen species (ROS) in NAFLD induces liver inflammation and damage, and melatonin has strong antioxidant effects. Therefore, melatonin supplementation could improve liver inflammation induced by ROS, and some studies are available, but this review does not mention about that at all. This review ignores critical roles of melatonin in liver pathology.
· In conclusion, the authors say as if current studies do not show anything, but there are multiple studies for the role or therapeutic effects of melatonin in NAFLD/NASH. Ref#35 Ghosh et al. 2023 or Sato et al. 2019 are previous reviews covering the field. Compared to these previous reviews, this review covers only generic information and does not show deep and latest studies.
· This review does not discuss about the therapeutic potentials of melatonin at all. There are some important studies, such as Pakravan et al. 2017 and Bahrami et al. 2020, showing the therapeutic effects of melatonin in human subjects. Overall, this review does not provide sufficient useful information to readers, and does not provide the merit for publication in the current format.
None.
Author Response
Point 1: According to the title, the authors seems to focus on the roles of melatonin in pathogenesis of NAFLD. Melatonin regulates circadian rhythms, and disrupted circadian rhythms are associated with lipid metabolism, fat deposition, and diabetes. Melatonin supplementation could improve pathogenesis via improving circadian rhythms. There are some studies showing that, but this review does not mention about this at all.
Response 1: From line 141/142 the following information on the role of melatonin as regulator of circadian rhythms was added:
Melatonin probably consolidate the whole rhythmic organisation by it’s chronobiotics properties [28, 29]. It is believed that through its prospective effects, melatonin can control the circadian time of the suprachiasmal nucleus, and pancreatic ß-cells and islets and the metabolically most important tissues (liver, muscle, and adipose tissue) are targets for melatonin [28]. As a strong chronobiotic, melatonin influences the circadian distribution of metabolic processes, synchronizing them to the activity-feeding/rest-fasting cycle [30]. Melatonin is an important player in regulation of energy balance and carbohydrate, and adipocyte metabolism [29, 30]. Table 1 summarizes the metabolic and chronobiological effects of melatonin that influence energy metabolism, and ultimately body weight.
Тable 1. Metabolic and chronobiological actions of melatonin [30]
Metabolic action |
Chronobiological action |
Regulates energy flow to and from storages |
Circadian synchronization of white adipose tissue metabolism |
Regulates energy expenditure |
Circadian synchronization of skeletal muscle metabolism |
Potentiates central and peripheral actions of insulin |
Circadian synchronization of insulin synthesis, secretion and action |
Regulates glycemia and lipidemia |
Circadian synchronization of hepatic metabolism |
Since the regulating system of melatonin secretion is complex, there are many pathological situations, where melatonin secretion can be disturbed [20]. For example, with aging, night shift work or illuminated environments during the night, the secretion of melatonin is suppressed [30]. As a result, consequences associated with disturbances in metabolism (insulin resistance, glucose intolerance, sleep disturbance, dyslipidemia) and in metabolic circadian synchronization (chronodisruption) occur, which can lead to metabolic disorders and obesity. It is assumed that the use of melatonin replacement therapy may protect and/or help eliminate these pathologies. The effects of melatonin administration (in experimental or clinical studies, in treatment) depend on a number of factors such as time and route of administration, concentration and duration of administration, regularity of intakes, specificity of the target organ (for example presence or absence of different melatonin receptors) [28]. Ectopic fat accumulation, particularly in the liver, is frequently observed in obese persons and is strongly associated with metabolic dysfunction, including multiorgan insulin resistance and dyslipidemia [31]. The possibility of interaction between exercise and diet on the mechanisms that regulate liver fat accumulation and depletion is discussed. Obesity is a multifactorial disease (it's not just increased food intake that matters), leading to difficulties in correcting metabolic disorders [29]. One population-based propensity score-matched study evaluate the association between circadian misalignment and MAFLD and found that prevalence of MAFLD was higher in the circadian misalignment group than the non-circadian misalignment group (45,41% vs 28.41%, P<0.001) [32]. The data also suggest that presence of circadian misalignment increased the risk of MAFLD by more than twofold and that circadian misalignment is independently associated with MAFLD, while short sleep duration alone (< 6 hours) is not independently associated with this risk. The authors conclude that in addition to diet, exercise and pharmacological therapy, it is appropriate to make efforts to improve existing sleep or chronotype disorders. In a study by Wyatt et al. [33] oral melatonin (0.3 mg or 5.0 mg) or identical-appearing placebo was administered 30 minutes prior to each sleep episode during forced desynchrony in 36 healthy men and women, between the ages of 18 and 30. According to survey data, administration of exogenous melatonin in young men and women has circadian phase-dependent hypnotic properties that enhance sleep consolidation occurring outside the period of endogenous melatonin secretion. Results support the hypothesis that both exogenous and endogenous melatonin attenuate the wake-promoting drive from the circadian system. The action of melatonin as a chronobiotic was confirmed by a study in seven totally blind people with free-runing circadian rhythms, who received 10 mg of oral melatonin or placebo nightly, one hour before their preferrable bedtime [34]. The results indicate that the phaseadvancing effects of melatonin are of sufficient magnitude to entrain free-running circadian rhythms in most blind persons who have such rhythms, thereby preventing severe sleep disturbance. In a mini review of D'Souza et al. [35] the pathogenesis of NAFLD in the context of sleep and circadian abnormalities is explored. The relationship between various sleep disorders (obstructive briefs apnea, circadian rhythm disturbances, insufficient sleep) and NAFLD was analyzed. Effective non-pharmacological therapeutic options, such as lifestyle modification through diet and proper exercise regimen, are discussed. Promising options for pharmacological therapy are melatonin, vitamin E, thiazolidinediones, and fecal microbiota transplantation. According to Lin et al. [36] nocturnal hypoxia in patients with obstructive sleep apnea might be a risk factor in the progression of NAFLD. The intermittent hypoxia seen in obstructive sleep apnea may contribute to fibrotic changes in the liver [35].
Point 2: Reactive oxygen species (ROS) in NAFLD induces liver inflammation and damage, and melatonin has strong antioxidant effects. Therefore, melatonin supplementation could improve liver inflammation induced by ROS, and some studies are available, but this review does not mention about that at all. This review ignores critical roles of melatonin in liver pathology.
Response 2: Melatonin is important not only as a chronobiotic, but also plays a role as an antioxidant. The numerous actions of melatonin as a direct and indirect antioxidant are discussed by Reiter et al. in 2003 [37]. Melatonin acts as an antioxidant by directly scavenging free radicals, stimulating antioxidant enzymes, increasing the efficacy of mitochondrial oxidative phosphorylation and reducing electron leakage, increasing the efficacy of other antioxidants. The mechanisms by which melatonin affects oxidative stress have been studied at the cellular and tissue level (liver, brain, kidney) [38, 39]. In laboratory conditions, a gastro- and hepatoprotective effect of exogenous melatonin has been established [40, 41]. The results suggest that melatonin ameliorates carbon tetrachloride-induced hepatic fibrogenesis in rats via inhibition of oxidative stress and proinflamatory cytokines production [41]. The rats injected subcutaneously with carbon tetrachloride for 6 weeks resulted in hepatic fibrotic changes increased hydroxyproline and malondialdehyde (MDA) levels, and decreased glutathione peroxidase and superoxide dismutase levels, whereas melatonin reversed these effects. Melatonin inhibited the expression of nuclear factor-kappa B in liver tissue and decreasing production of proinflammatory cytokines such as tumor necrosis factor-ɑ (TNF-ɑ) and interleucin-1ß (IL-1ß) from Kupffer cells in fibrotic rats. The effect of melatonin on the treatment of patients with NAFLD was evaluated in a study of Bahrami et al. [42]. This randomised double-blind, placebo-controlled study showed that administration of 6 mg/day melatonin for 12 weeks improved a number of factors associated with NAFLD such as enzymes (aspartate aminotransferase, AST; alanine aminotransferase, ALT), anthropometric factors (weight, waist circumference, abdominal circumference), blood pressure, serum leptin levels and the grade of fatty liver.
Point 3: In conclusion, the authors say as if current studies do not show anything, but there are multiple studies for the role or therapeutic effects of melatonin in NAFLD/NASH. Ref#35 Ghosh et al. 2023 or Sato et al. 2019 are previous reviews covering the field. Compared to these previous reviews, this review covers only generic information and does not show deep and latest studies.
Response 3:
Conclusions: The currently existing hypotheses about the pathogenesis of NAFLD do not provide a complete picture of the mechanisms underlying the disease and the causes of its progression. A review of experimental study data on the role of melatonin in the pathogenesis of NAFLD shows that melatonin executes its effects by influencing various signaling pathways. Melatonin administration improves mitochondrial function, reduces inflammatory markers and the formation of reactive oxygen species, restored levels of antioxidant factors, improve liver steatosis and fibrosis in high fat diet-induced NAFLD and NASH. Clinical studies have shown that exogenous melatonin lowers hepatic aminotransferases and gamma-glutamyl transpeptidase and lipid profile indicators and it is assumed that melatonin should be practically important in prevention of the progression of liver damage in NAFLD and NASH. In addition to exogenous melatonin, melatonin agonists are used in clinical practice (Ramelteon, Agomelatine, Tazimelteon, prolonged-release melatonin Circadin), approved for the treatment of insomnia, depression and disturbances of sleep-wake circadian rhythms. In connection with evidence of liver function injury when using agomelaton, frequent monitoring of liver function is recommended. The use of melatonin receptor agonist ACH-000143 for the treatment of NAFLD is under investigation. In a number of surgical interventions, melatonin has been used for premedication. As side effects of melatonin administration, single cases of headache, dizziness, nausea have been reported. The diversity of doses and route of administration of exogenous melatonin in different studies is noteworthy. Melatonin is found in a number of foods (vegetables, fruit, nuts, meat) in different concentrations. There is evidence to show that eating foods rich in melatonin increases serum melatonin, but it's not yet clear how dietary melatonin consumption affects the melatonin profile in humans.
Point 4: This review does not discuss about the therapeutic potentials of melatonin at all. There are some important studies, such as Pakravan et al. 2017 and Bahrami et al. 2020, showing the therapeutic effects of melatonin in human subjects. Overall, this review does not provide sufficient useful information to readers, and does not provide the merit for publication in the current format.
Response 4:
Therapeutic potentials of melatonin in NAFLD
In 2019, Baiocchi et al. [87] express the opinion that all studies related to the potential therapeutic effects of melatonin on NASH in rodents and humans did not pinpoint the possible molecular mechanisms by which melatonin protects against NASH, but rather, focused only on the general antioxidant and cytoprotective properties of melatonin in this setting. Two years earlier, Zang et al. [88] analyzed the protective effect of melatonin on liver injuries induced by various factors and liver diseases such as liver steatosis, non-alcohol fatty liver, hepatitis, liver fibrosis, liver cirrhosis, and hepatocarcinoma. Our review of experimental data on the role of melatonin in the pathogenesis of NAFLD shows that melatonin affects a number of signaling pathways, resulting in improved inflammation, oxidative stress, lipid and fat metabolism, improved mitochondrial physiology. Аccording to Sato et al. [89] melatonin has potentials for novel treatments of liver diseases by decreasing oxidative stress or restoring circadian rhythms and functions. However, related studies of melatonin applied to clinical treatment for liver injuries and diseases are limited [88]. Mohammadi et al. [90] administered melatonin (10 mg/day), metformin (500mg/day) and vitamin E (800IU/day) to patients with NAFLD for six months and found that melatonin reduced serum aminotransferases, triglycerides, cholesterol and fasting glucose when comparing these parameters before and after medication with melatonin. When comparing these indicators against a control group (received plasibo), only low density lipoprotein and AST had significant changes. Based on the improvements shown by ultrasonography, the greatest improvement was demonstrated by metformin and the autors concluds that metformin is a better chiose for the treatment of these patients. Mohammadi et al. [90] suggest that melatonin be considered an effective treatment of NAFLD as this drug made improvement in different aspects of NAFLD injuries. Gonciarz et al. [91] evaluated the effects of 24 weeks of lifestyle intervention combined with 10 mg/day melatonin treatment (5 mg at 09:00 h and 5 mg at 21:00 h) on plasma liver enzymes levels AST, ALT, gamma-glutamyl transpeptidase (GGT), alkaline phosphatase (ALP)], concentrations of lipids (total cholesterol, triglycerides), glucose and melatonin in 30 patients with NASH. A control group of 12 patients with NASH who received placebo was used for comparison. The study demonstrates that AST and GGT levels decrease significantly only in melatonin-treated group. The decrease in median plasma ALT level in melatonin-treated group at week 18 and week 24 was significantly more intense (P<0.5) than that observed in control group, however at follow-up the difference between the two groups was not significant. The higher ALT, AST and GGT levels shown at follow-up in comparison with that found at 18th and 24th week of treatment reflected the high efficacy of melatonin, linked closely to the period of medicine administration. Plasma concentration of melatonin (pg/ml) in melatonin-treated group averaged 7.5±3.5 at baseline and increased to 52.5±17.5 at 24th week, and no patients complained somnolence and no significant side-effects were observed. Cichoz-Lach et al. [92] evaluate the effects of melatonin and L-tryptophan on selected biochemical parameters and proinflammatory cytokines of blood in 45 patients with NASH divided into three groups: the first group received preparation Essentiale forte 3 times a day and L-tryptophan 500 mg twice a day; the second group received Essentiale forte in the above doses and melatonin 5 mg twice a day; the third group received only Essentiale forte three times a day. Treatment lasted 4 weeks. In all participants plasma biochemical parameters (ALT, AST, ALP, GGT, bilirubin, total cholesterol, triglycerides, LDL, HDL) and cytokines (IL-1, IL-6 and TNF-ɑ) were measured after 4 weeks of treatment and were compared with the results evaluated at the start of the study. The study showed that the addition of melatonin or its precursor, L-tryptophan to Essentiale forte therapy resulted in a statistically significant decrease in plasma levels of key pro-inflammatory cytokines such as IL-1, IL-6 and TNF-ɑ. This effect can be explained by the antioxidant action of melatonin and leads to an improvement in the therapy of NASH. The beneficial effect is also accompanied by a decrease in GGT and triglycerides levels. Based on the results obtained these researchers suggests that treatment with melatonin should be practically very important in prevention of the progression of liver damage in NAFLD and NASH. A similar design has the study of Celinski et al. [93] which also determine the effects of tryptophan and melatonin on the biochemical parameters in patients with NAFLD. In addition it is evaluate the effects of tryptophan and melatonin in improvement of liver tissue in selected NAFLD patients (n = 9) after 14-months of treatment period. Significant reduced activity of GGT and values of triglycerides, LDL-cholesterol, IL-1, IL-6 and TNF-ɑ were found in the groups received melatonin and tryptophan, compared to group received only Essentiale forte. The study findings demonstrate that melatonin and tryptophan substantially reduce the levels of pro-inflammatory cytokines and improve some parameters of fat metabolism in patients with NAFLD. In few patients with NASH melatonin and tryptophan reduced the inflammation in liver. It has been concluded that melatonin is worth considering for the therapy of NAFLD, especially in patients with impaired fat metabolism (hypertriglyceridemia and hyper-LDL cholesterolemia). No side effects of melatonin and tryptophan were observed; for instance none of patients complained on excessive sleepiness and/or dizzines. Pakravan et al. [94] studied the effect of melatonin in 100 patients with NAFLD aged 22 to 65 years, divided into two groups: a case group (n=50) received melatonin tablets twice a day for 6 weeks, and a control group (n=50) received twice daily placebo for the same period of time. During the study, the patients followed the same diet and exercise regime. Results showed that in case group the mean of weight, waist, systolic and diastolic blood pressure, high sensitive C-reactive protein and ALT after treatment was significantly decreased compare to baseline, also, melatonin significantly decrease diastolic blood pressure, AST, and high sensitive C-reactive protein in case group more than control group. In addition, in most of the patients who received melatonin grade of fatty liver improved than controls. These results demonstrated that use of melatonin in patients with NAFLD was more affected than placebo with no serious side effects. Melatonin significantly decrease liver enzymes in cases than placebo, therefore, the use of melatonin in patients with NAFLD can be effective.
It is hypothesized that new compounds that act as specific melatonin agonists or antagonists will contribute to a better understanding of melatonin's mechanism of action [29]. Melatonin analogues (agonists and antagonists) differ in their chemical structure and affinity for melatonin receptors [95]. Currently powerful, lipophilic, non-selective MT1/MT2 high exposure agonists in the brain, such as Ramelteon, Agomelatine, Tazimelteon, prolonged-release melatonin (Circadin) are approved for treatment of insomnia, depression and circadian rhythms sleep-wake disorders [96]. In a systematic review Freiesleben at Furczyk [97] evaluate the potential risk posed by agomelatine as an antidepressant of inducing liver injury. Agomelatine was found to be associated with higher rates of liver injury than both placebo and the four active comparator antidepressants used in the clinical trials for agomelatine, with rates as high as 4.6% for agomelatine compared to 2.1% for placebo, 1.4% for escitalopram, 0.6% for paroxetine, 0.4% for fluoxetine, and 0% for sertraline. The review also provides evidence for the existence of a positive relationship between agomelatine dose and liver injury. According to researchers it is essential that clinicians continue to monitor liver function frequently, as prescribed by the manufacturer of agomelatine. Early detection, followed by best practice treatment plan reactions (e.g., treatment discontinuation), remain the most efficient responses toward possible manifestations of liver damage. Ferreira et al. [98] reported the discovery of a new powerful melatonin receptor agonist, benzoimidazole derivative compound 10b, that reduced weight gain, liver triglycerides and steatosis in HFD rats. Two-month oral administration of 10b in high-fat diet rats led to a reduction in body weight gain with superior results on hepatic steatosis and triglyceride levels. An early toxicological assessment indicated that 10b (also codified as ACH-000143) was devoid of genotoxicity, and behavioral alterations at doses up to 100 mg/kg p.o. Based on its efficacy, oral pharmacokinetics and safety, compound 10b, are selected for further investigation as a candidate drug against NAFLD/NASH.
Round 2
Reviewer 2 Report
In this revised version, the content I commented on last time is firmly included. I think that this version is acceptable.
Reviewer 3 Report
No further comments
only minor